# Gentamicin Targets Acid Sphingomyelinase in Cancer: The Case of the Human Gastric Cancer NCI-N87 Cells

**DOI:** 10.3390/ijms20184375

**Published:** 2019-09-06

**Authors:** Elisabetta Albi, Samuela Cataldi, Maria Rachele Ceccarini, Carmela Conte, Ivana Ferri, Katia Fettucciari, Federica Filomena Patria, Tommaso Beccari, Michela Codini

**Affiliations:** 1Department of Pharmaceutical Sciences, University of Perugia, 06126 Perugia, Italy (E.A.) (S.C.) (M.R.C.) (C.C.) (F.F.P.) (T.B.); 2Institute of Pathologic Anatomy and Histology, University of Perugia, 06126 Perugia, Italy; 3Department of Experimental Medicine, University of Perugia, 06100 Perugia, Italy

**Keywords:** lipid, sphingomyelin, sphingomyelinase, cancer, drug target

## Abstract

Emerging literature implicates acid sphingomyelinase in tumor sensitivity/resistance to anticancer treatments. Gentamicin is a drug commonly used as an antimicrobial but its serendipity effects have been shown. Even though many evidences on the role of gentamicin in cancer have been reported, its mechanism of action is poorly understood. Here, we explored acid sphingomyelinase as a possible new target of gentamicin in cancer. Since gastric cancer is one of the most common cancers and represents the second cause of death in the world, we performed the study in NCI-N87 gastric cancer cell line. The effect of the drug resulted in the inhibition of cell proliferation, including a reduction of cell number and viability, in the decrease of MIB-1 proliferative index as well as in the upregulation of cyclin-dependent kinase inhibitor 1A and 1B (*CDKN1A* and *CDKN1B*), and growth arrest and DNA-damage 45A (*GADD45A*) genes. The cytotoxicity was apoptotic as shown by FACS analysis. Additionally, gentamicin reduced HER2 protein, indicating a minor tumor aggressiveness. To further define the involvement of sphingomyelin metabolism in the response to the drug, gene and protein expression of acid and neutral sphingomeylinase was analyzed in comparison with phosphatase and tensin homolog deleted on chromosome 10 (PTEN) and vitamin D receptor (VDR), molecules involved in cancer. Gentamicin induced a downregulation of *PTEN*, *VDR*, and neutral sphingomyelinase and a strong upregulation of acid sphingomyelinase. Of note, we identified the same upregulation of acid sphingomyelinase upon gentamicin treatment in other cancer cells and not in normal cells. These findings provide new insights into acid sphingomyelinase as therapeutic target, reinforcing studies on the potential role of gentamicin in anticancer therapy.

## 1. Introduction

Sphingomyelin (SM) is a bioactive sphingolipid recognized as an important signaling molecule in cell proliferation, differentiation, apoptosis, and cancer [1]. SM metabolism is a multifaceted network that involves many enzymes responsible for SM and phosphatidylcholine (PC) balance by producing second mediators such as ceramide and diacylglycerol (DAG). Both these molecules may elicit several opposite effects within the cell [2]. SM is hydrolyzed by sphingomyelinase (SMase) to produce phosphocholine and ceramide, and it is synthesized by SM-synthase by using phosphocholine from PC producing DAG. PC is restored by using SM as a source of phosphocholine [2]. The SM breakdown is carried out by different SMase isoenzymes belonging to three families with distinct and often opposite roles: neutral sphingomyelinase (nSMase), acid sphingomyelinase (aSMase), and alkaline sphingomyelinase (alkSMase) [3]. Studies from several laboratories identified nSMase as a possible tumor suppressor. nSMase overexpression delayed cell growth in MCF-7 breast cancer cells [4], where it was considered a target for daunorubicin [5]. In addition, nSMase overexpression reduced hepatocyte proliferation [6], and its deficiency led to spontaneous liver tumor [7]. Moreover, nSMase gene mutations were identified in acute myeloid and lymphoid leukemias [8]. There is an emerging body of literature implicating aSMase in the modulation of cancer progression [9], contributing to the apoptosis of tumor cells and determining tumor sensitivity/resistance to anticancer treatments [10]. Perrotta et al. (2015) highlighted the existence of an aSMase-autophagy axis whose imbalance plays a role in cancer [11].

Gentamicin (GM) is an aminoglycoside molecule primarily used as a gram-negative antimicrobial drug due to its ability to bind and cleave the 30S subunit of the bacterial ribosome [12]. Previous studies revealed a nephrotoxic action for high doses of GM with particular focus on both apoptosis [13] and necrosis [14] of tubular epithelial cells. These effects were mediated by nitric oxide production [15] and by SMase inhibition [16]. In the last ten years, the serendipity function of GM has been elucidated in cancer. In fact, GM has been implicated in the onset and progression of non-Hodgkin’s T cell human lymphoblast lymphoma (SUP-T1) by regulating SM metabolism [17] and in increasing the efficacy of chemotherapy treatment [18].

The above data provide strong evidence that nSMase and aSMase are important regulators of cancer cell fate. Starting from these findings, we studied for the first time, the effect of GM treatment on nSMase and aSMase in NCI-N87 gastric cancer cell line, by demonstrating that the drug delays cancer cell proliferation and aggressiveness by involving specifically aSMase.

## 2. Results

### 2.1. Advances in Anticancer Action of Gentamicin

GM has been described to be able to induce cancer cell death. In this study, we evaluated whether GM can suppress the human gastric cancer cell growth. A dose response of GM revealed that with increasing dose from 0.75 to 1.5 mM for 24 h, a significant reduction of NCI-N87 cell number was observed (Figure 1a). However, considering all the concentrations studied from 0.25 to 20 mM, the dose–response effect is not linear, as it happens for other drugs or in other experimental models [17,19,20]. The results showed that high doses did not induce reduction of cell growth similar to the effect obtained with very low doses (Figure 1a). To determine the effect of GM on cell death, a trypan blue assay was performed. No significant difference in cell death was induced by GM treatment from 0.25 to 2 mM at 24 h (Figure 1a). Conversely, a 72 h time point showed that the long time treatment significantly decreased the cell number, specially by 1.75 and 2 mM GM treatment. At the same concentrations, accumulated cell death was shown (Figure 1a). In addition, after 24 h from treatment, cell viability, assayed by the MTT test, did not change with concentration up to 0.5 mM GM, but decreased progressively with concentrations between 0.75 and 2 mM (Figure 1b). Therefore, the results indicated that the cells treated with GM were numerically similar to the control, but their viability was reduced. Cell viability following exposure to 1.5 mM GM was around 70%. Based on these results, 1.5 mM GM concentration was chosen for the following experiments. We then set out to determine whether GM-induced cytotoxicity was apoptotic. Therefore, we performed FACS analysis after 24 h from 1.5 mM GM treatment and we compared the results with those of the control sample. As shown in Figure 1c, there was an increase of apoptosis after GM treatment (control sample 5.21% ± 0.25%; GM treated sample 7.61% ± 0.16%) although there were no variations from the count of dead cells, because the last is a more approximate technique. In addition, FACS analysis showed different distribution of the control and experimental cells in all phases of the cell cycle. In fact, control cells were 65.4% ± 1.2% in G0/G1, 28.32% ± 0.8% in S, and 6.25% ± 0.5% in G2/M phase of the cell cycle, and GM treated cells were 70.17% ± 1.41% in G0/G1, 25.14% ± 0.36% in S, and 4.67% ± 0.14% in G2/M phase of the cell cycle, by indicating an accumulation of the cells in G0/G1 phase of the cell cycle.

These data were consistent with the upregulation of *CDKN1A* and *CDKN1B* cyclin-dependent kinase inhibitor genes (Figure 1d). Hematoxylin–eosin staining revealed that NCI-N87 GM-treated cells exhibited large size, suggesting that GM induced a change of cell morphology (Figure 2a). Immunohistochemistry analysis by using Ki-67 (MIB-1) as proliferation marker revealed that GM treatment caused a reduction of cell labeling, confirming a significant inhibition of cell growth [21] (Figure 2b). Furthermore, the HercepTest, a semi-quantitative immunohistochemistry assay, was performed to determine the expression of HER2 protein, a transmembrane tyrosine kinase receptor that plays a key role in the development and progression of gastric cancer cells [22]. Images showed a strong reduction of labeling in the GM-treated cells (Figure 2c,d). Densitometric analysis (Figure 2e). In accordance, the growth arrest and DNA-damage 45A (*GADD45A*) gene was strongly upregulated (Figure 2f). Taken together, these results show the potential role of GM in the reduction of gastric cancer cell growth.

### 2.2. Gentamicin Alters Sphingomyelin Metabolism

To investigate whether GM-induced cell growth inhibition was linked to the change of SM metabolism, the gene expression of nSMase and aSMase was measured. As shown in Figure 3, the treatment with 1.5 mM GM for 24 h induced a downregulation of nSMase and approximately a threefold increase in aSMase gene expression. To verify a specific SM metabolism enzyme deregulation after GM treatment, other genes involved in cancer, such as vitamin D receptor (*VDR*) [17], importin 7 (*Ipo7*) [22], and anti-phosphatase and tensin homolog deleted on chromosome 10 (*PTEN*) [17], were considered. Figure 3 shows no changes of *Ipo7* expression and a significant downregulation of *VDR* in accordance with our previous results [17]. Contrary to expectations, a downregulation of the *PTEN* gene was obtained (Figure 3). Then we performed immunoblotting analysis to study nSMase, aSMase, Ipo7, VDR, and PTEN protein expression. Figure 4 provides strong evidence that proteins changed according to the gene expression. Results suggested that GM-induced reduction of cancer cell growth was not mediated by the PTEN pathway. To completely exclude the involvement of PTEN, AKT and phosho-AKT (p-AKT) have been studied. Figure 5 illustrates how AKT and p-AKT were not modified following treatment of cells with GM.

Based on these results, we could hypothesize that aSMase might specifically mediate GM response in gastric cancer cells. To investigate whether aSMase could be a specific target of GM in cancer cells, the experiment was repeated by incubating normal cells such as thyrocytes (FRTL-5), embryonic hippocampal cells (HN9.10), and lymphocytes, and other cancer cells as and lymphoma cells (SUP-T1) and hepatoma cells (H35), with 1.5 mM GM for 24 h. Interestingly, GM specifically upregulated the aSMase gene only in cancer cells (Figure 6). The analysis of aSMase protein supported this results (Figure 7). Therefore, these results can induce to hypothesize that aSMase could be a specific target of GM in cancer cells.

## 3. Discussion

The aim of the study was to demonstrate that SM metabolism enzymes could be potential targets of GM. To this end, we used human gastric cancer cells since it is one of the most common cancers and represents the second cause of death in the world, in order to highlight a new serendipity function of the drug to be added to those already reported in the literature. In fact, GM, a known antibacterial agent, induced apoptosis, necrosis, and cancer in mammalians [13,14,16,18]. Nowadays, the treatment of gastric cancer is primarily based on surgical resection and different chemotherapy protocols [23,24,25]. A recent review highlighted the importance of monoclonal antibody use [26]. Interestingly, Lei et al. (2017) demonstrated that quercetin, a flavonoid widely present in vegetables and fruits, potentiated the efficacy of anticancer drugs [27]. Recently, GM has been described as a read-through agent for the treatment of rectal cancer [28]. This is the first study showing the action of GM in human gastric cancer NCI-N87 cells. We found that GM treatment delayed cell growth by inducing a slight upregulation of *CDKN1A* and *CDKN1B* and a strong upregulation of *GADD45A* gene expression. It is well known that CDKN1A, a kinase induced in response to DNA damage, mediates cell cycle arrest in G1 and G2 phases [29]. Moreover, CDKN1B plays a critical role in the regulation of G1/S transition of the cell cycle [30] and GADD45A is involved in promoting cell death [31]. Although a wide literature indicated that the tumor suppressor *PTEN* was required for the anticancer action of both drug [32] and natural product [33], we found no evidence about the involvement of the PTEN/AKT pathway as a direct substrate of GM. Moreover, since VDR gene and protein expression levels were slightly reduced we suggest that the mechanism underlying the action of GM did not depend on VDR expression. There are conflicting findings regarding the impact of VDR in cancer. Matusiak et al. (2005) found that VDR protein levels decreased in relation with colon cancer cell de-differentiation [34]. Recently, Liu et al., (2018) demonstrated that *VDR* was a target for miR-1204 to promote breast cancer cell proliferation, tumorigenesis, and metastasis [35]. These controversial observations might be due to the deregulation of vitamin D metabolism in many types of cancer [36]. Interestingly, we revealed modifications in the SM pathway through a slight reduction of nSMase and a strong increase of aSMase gene and protein expression. The specificity of GM action on aSMase in cancer cells was demonstrated showing an increase of aSMase gene and protein expression in other cancer cell lines such as SUP-T1 (lymphoma cells) and H35 (hepatoma cells). It was very relevant considering that nSMase is correlated with cell growth [37] and aSMase with autophagy and/or apoptosis [11]. The definitive proof that aSMase is a GM target could be obtained by testing the effect of aSMase knockdown by siRNA transfection. However, it represents the cellular model that reproduces Niemann–Pick disease in vitro. The lack of the enzyme causes an accumulation of SM that induces cellular degeneration and death with a great experimental variability, and, therefore, it is not easy to establish the effect of a GM in these cells.

Our results suggest that aSMase can be considered a potential target of GM in cancer cells. Our attention focused on gastric cancer considering that it is the most common cancer and represents the second cause of death in the world. In this study, we provide evidence for anti-gastric cancer properties of an anti-microbial drug, indicating a serendipitous finding of GM that acts via aSMase. Thus, GM could be a valuable aid for anticancer therapy at low costs and with reduced collateral effects.

## 4. Materials and Methods

### 4.1. Materials

Human gastric cancer NCI-N87 cell line was purchased from Istituto Zooprofilattico Sperimentale della Lombardia e dell’Emilia Romagna ‘Bruno Ubertini’ (Brescia, Italy). Thyroid epithelial FRTL-5 cells were prepared and characterized as previously reported [38]. Lymphocytes were from three donors (Centro Trasfusionale, Ospedale Silvestrini, Perugia, Italy), as previously reported [17]. Non-Hodgkin’s T-cell human lymphoblastic lymphoma (SUP-T1) were from Biological Materials Bank (ICLC) CBA, Genoa, Italy. H35 hepatoma cells were obtained from the European Collection of Animal Cell Cultures (Salisbury, UK). RPMI 1640, L-glutamine, trypsin, and ethylenediaminetetraacetic acid disodium and tetrasodium salt (EDTA) were from Microtech Srl (Pozzuoli, NA, Italy). Fetal bovine serum (FBS), penicillin–streptomycin, Dulbecco’s phosphate buffered saline pH 7.4 (PBS), High-Capacity cDNA Reverse Transcription Kit, TaqMan^®^Gene Expression Master Mix, and all gene expression TaqMan assays used were from Thermo Fisher Scientific (Waltham, MA, USA). Dimethyl sulfoxide (DMSO) was purchased from Carlo Erba Reagents Srl (Milan, Italy). 3-[4,5-Dimethyl-2-thiazolyl]-2,5-diphenyl-2-tetrazoliumbromide (MTT) and gentamicin sulfate salt (477.6 molecular weight) were purchased from Sigma-Aldrich Srl (St. Louis, MO, USA). RNAqueous^®^-4PCR kit was from Ambion Inc. (Austin, TX, USA). Anti-aSMase, anti-nSMase, anti-vitamin D receptor (VDR), anti-importin 7 (Ipo7), and anti-phosphatase and tensin homolog deleted on chromosome 10 (PTEN) were from Abcam (Cambridge, UK). Anti-AKT and phosphor-AKT (p-AKT) were from Cell Signaling Technology—EuroClone (Milano, Italy). Horseradish peroxidase-conjugated goat anti-rabbit secondary antibodies were from Santa Cruz Biotechnology (Dallas, TX, USA).

### 4.2. Cell Culture

NCI-N87 cells were grown as previously reported [39]. Cells were cultured in RPMI 1640 medium containing 2 mM L-glutamine, 100 U/mL penicillin, and 100 μg/mL streptomycin in the presence of 10% FBS. Thyroid epithelial FRTL-5 cells were grown in Ham’s modified F-12 with 5% calf serum and six hormones: 10 ng/mL glycyl-l-histidyl-l-lysine acetate (Sigma), 10^−8^ M hydrocortisone (Sigma), 10 µg/mL insulin (Sigma), 10 µg/mL somatostatin (Sigma), 5 µg/mL transferrin (Sigma), and 10 mU/mL TSH (Sigma), as previously reported [38]. Lymphocytes were extracted from the peripheral blood by using “Lymphocyte Separation Medium” according to the protocol instructions for use (Lonza Group, Basel, Switzerland) and cultured in RPMI 1640 medium with penicillin, streptomycin, and amphotericin B added in the presence of 10% FCS, as previously reported [17]. SUP-T1 cells were cultured in DMEM supplemented with 10% fetal bovine serum, 2 mM L-glutamine, 100 IU/mL penicillin, 100 g/mL streptomycin, and 2.5 g/mL amphotericin B (fungizone), as previously reported [39]. H35 hepatoma cells were seeded in 25 cm^2^ flasks and were grown in monolayer in DMEM enriched with 10% FBS, 2 mM of L-glutamine, 100 IU/mL of penicillin, 100 μg/mL of streptomycin, and 250 μg/mL of amphotericin B, as previously reported [40]. All cells were maintained at 37 °C in 5% of CO_2_ and 95% humidity.

### 4.3. GM Dose-Dependent Effect

To establish the GM dose-dependent effects in NCI-N87 cells, increasing concentrations of the drug from 0.25 to 2.0 mM were added to the culture medium for 24 h or 72 h. Cell death was counted by using a trypan blue dye exclusion assay and the viability was analyzed by MTT assay as previously reported [41].

### 4.4. Flow Cytometry Analysis

Flow cytometry analysis was performed and analyzed as previously reported [39]. Briefly, cells were collected, washed, and resuspended in 1 mL of hypotonic propidium iodide (PI) solution (50 μg mL^−1^ in 0.1% sodium citrate plus 0.1% Triton X-100; Sigma). The samples were placed for 1 h in the dark at 4 °C, and the PI fluorescence of individual nuclei was measured using an EPICS XL-MCL™ flow cytometer (Beckman Coulter, Inc., Miami, FL, USA). Apoptosis data were processed by an Intercomp computer and analyzed with EXPO32 software (Beckman Coulter). The cell cycle was analyzed by measuring DNA-bound PI fluorescence in the orange–red fluorescence channel (FL2) through a 585/42 nm bandpass filter with linear amplification. Analysis of distribution profiles was performed with ModFit LT software (Verity Software House, Topsham, ME, USA) to determine fractions of the population in each phase of the cell cycle (G0/G1, S, G2/M). At least 15,000 events were collected for each sample. Cells were gated on FL2-area versus FL2-width plots to exclude aggregates and debris from the analysis, as previously reported [39].

### 4.5. Morphological and Immunohistochemistry Analysis

NCI-N87 cells were incubated for 24 h in the presence or absence of 1.5 mM GM, and then were fixed in 96% ethanol for 5 min, included in paraffin and sectioned into 4-µm-thick sections as previously reported [42]. Hematoxylin–eosin staining, immunohistochemistry determination of Ki-67 (MIB-1 clone) and HER2 (HercepTest) proteins was performed as previously reported [39]. The observations were performed by using inverted microscopy EUROMEX FE 2935 (Papenkamp 206836 BD Arnhem, The Netherlands) equipped with a CMEX 5000 camera system (40× magnification). The analysis of the immunostaining was performed by ImageFocus software.

### 4.6. Reverse Transcription Quantitative PCR (RTqPCR)

After 24 h of culture in the presence or absence of 1.5 mM GM, cells were used for total RNA extraction by using RNAqueous ^®^-4PCR kit and RT-PCR was performed by using Master Mix TaqMan^®^Gene Expression with 7.500 RT-PCR instrument (Applied Biosystems, Foster City, CA, USA). The following target genes were investigated: *SMPD1* (Hs03679347_g1), *SMPD4* (Hs04187047_g1), *VDR* (Hs00172113_m), *CDKN1A* (Hs 00355782_m1), *CDKN1B* (Hs 00153277_m1), *GADD45A* (Hs 00169255_m1), *IPO7* (Hs 00255188_m1), and *PTEN* (Hs 02621230_s1). *GAPDH* (Hs99999905_m1) was used as a housekeeping gene.

### 4.7. Western Blot

Western blot analysis was performed to determine protein expression levels in untreated and 1.5 mM GM-treated cells. Protein concentration was determined by Bradford assay [43]. Forty micrograms of proteins was loaded in a 10% SDS (sodium dodecyl sulfate)-polyacrylamide gel at 200 V for 60 min, transferred into 0.45 µm nitrocellulose membranes and blocked in 5% non-fat dry milk. The blot was incubated overnight at 4 °C with anti-aSMase, anti-nSMase, anti-VDR, anti-PTEN, anti-IPO7, anti-AKT, and anti-pAKT specific antibodies (1:1000), as previously reported [44].The blot was treated with horseradish peroxidase-conjugated goat anti-rabbit secondary antibodies (1:10,000). SuperSignal West Pico Chemiluminescent Substrate (Thermo Fisher Scientific) was used to detect chemiluminescent HRP substrate. The apparent molecular weight of proteins was calculated according to the migration of molecular size standards. The blot was stripped and treated for the analysis of β-tubulin used as the loading control. The band intensity was evaluated by densitometric analysis with the ImageJ program.

### 4.8. Statistical Analysis

Three independent experiments performed in duplicate were carried out for each analysis. Data are expressed as mean ± SD, Student’s *t* test was used for statistical analysis.

## Figures and Tables

**Figure 1 ijms-20-04375-f001:**
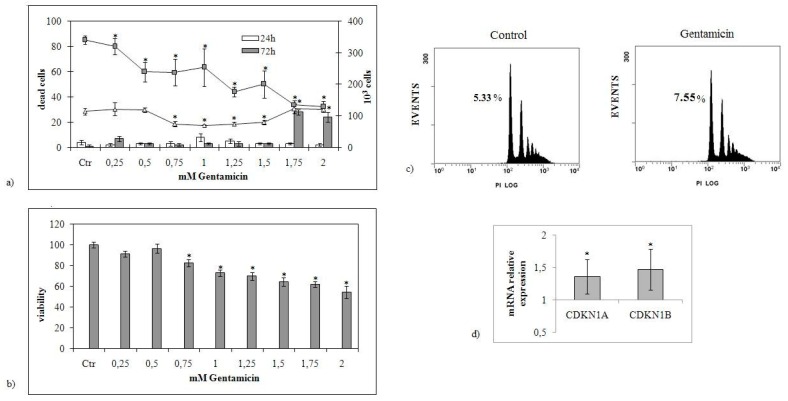
Effects of increasing doses of gentamicin on NCI-N87 cells. (**a**) The cells were counted 24 and 72 h after treatment with increasing concentrations of the drug (0.25–2.0 mM) (lines referred to the ordinate on the right). Cell death was evaluated by trypan blue staining (histograms referred to the ordinate on the left). Ctr, untreated sample. (**b**) Cell viability was assessed by MTT method and expressed as a percentage relative to that of the control cells set at 100%. (**c**) FACS analysis. (**d**) *CDKN1A, CDKN1B* gene expression, data are referred to the gentamicin (GM)-untreated sample (control) set at 1. *GAPDH* expression was used as a housekeeping gene. RT-PCR analysis was performed in control and experimental NCI-N87 cells collected 24 h after 1.5 mM GM treatment. The data represent the mean ± standard deviation (SD) of three independent experiments performed in duplicate (significance, * *p* < 0.001 versus control sample).

**Figure 2 ijms-20-04375-f002:**
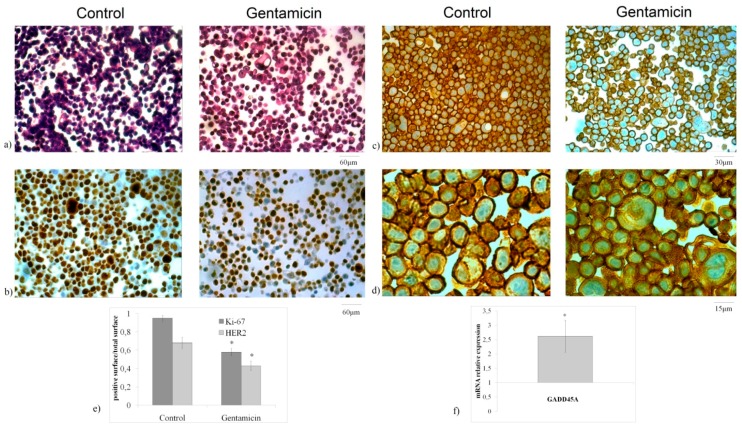
Effect of gentamicin after 24 h from treatment. (**a**) Cell morphology by hematoxylin–eosin staining; (**b**) MIB-1 immunohistochemistry staining; (**c**,**d**) HercepTest immunostaining. 10× magnification (**a**,**b**), 20× magnification (**c**), and 40× magnification (**d**); (**e**) densitometric analysis; (**f**) *GADD45A* gene expression, data are referred to the GM-untreated sample (control) set at 1. *GAPDH* expression was used as a housekeeping gene. RT-PCR analysis was performed in control and experimental NCI-N87 cells collected 24 h after 1.5 mM GM treatment. The data represent the mean ± SD of three independent experiments performed in duplicate (significance, * *p* < 0.001 versus control sample).

**Figure 3 ijms-20-04375-f003:**
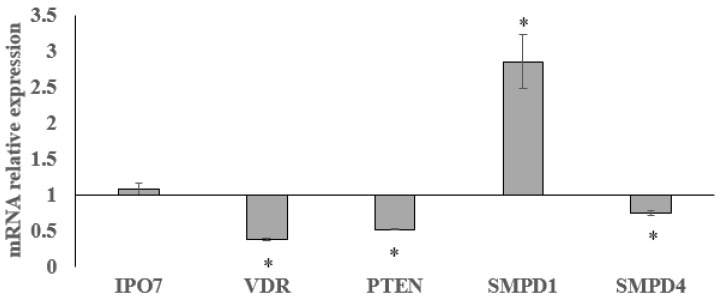
Effect of gentamicin on *Ipo7, VDR, PTEN, SMPD1, SMPD4*, gene expression. *GAPDH* expression was used as a housekeeping gene. RT-PCR analysis was performed in control and experimental NCI-N87 cells collected 24 h after 1.5 mM GM treatment. Data are expressed as the mean ± SD of three independent experiments performed in three PCR replicates (significance, * *p* < 0.001 versus control sample).

**Figure 4 ijms-20-04375-f004:**
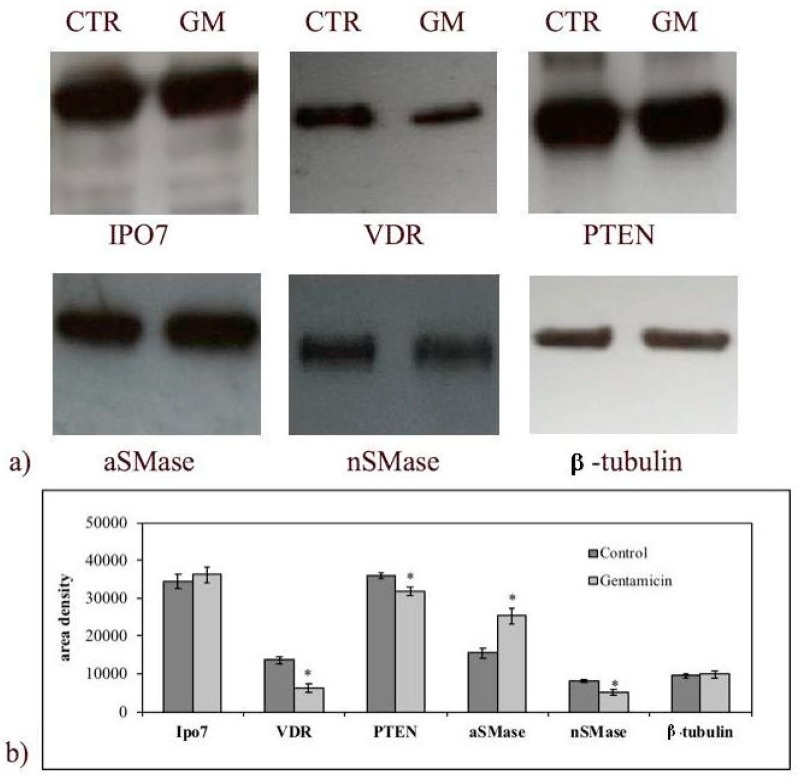
Effect of gentamicin on importin 7 (Ipo7), vitamin D receptor (VDR), phosphatase and tensin homolog deleted on chromosome 10 (PTEN), acid sphingomyelinase (aSMase), and neutral sphingomyelinase (nSMase) protein expression. Experiments were performed in control and experimental NCI-N87 cells collected 24 h after 1.5 mM GM treatment. (**a**) Immunoblots of proteins were probed with anti-IPO7, anti-VDR, anti-PTEN, anti-aSMase, and anti-nSMase and visualized by enhanced chemiluminescence (ECL). β-tubulin was used as loading control. (**b**) The area intensity was evaluated by densitometry scanning and analysis with ImageJ program, the data represent the mean ± SD of three experiments performed in duplicate (significance, * *p* < 0.001 versus control sample).

**Figure 5 ijms-20-04375-f005:**
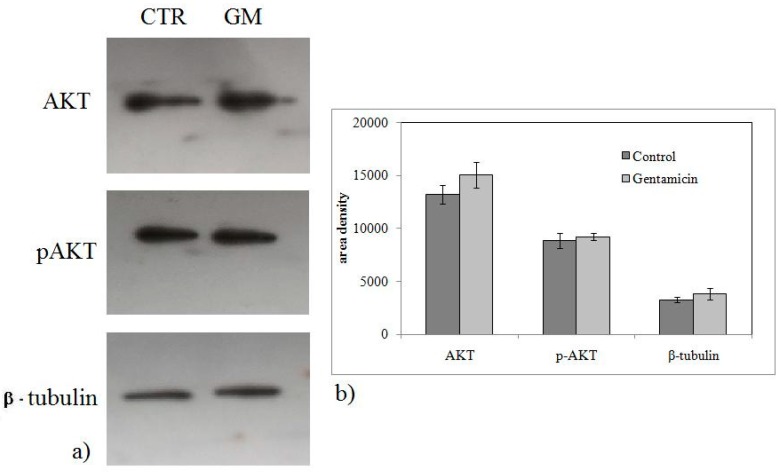
Effect of gentamicin on AKT/p-AKP protein expression. Experiments were performed in control and experimental NCI-N87 cells collected 24 h after 1.5 mM GM treatment. (**a**) Immunoblots of proteins were probed with anti-AKT and p-AKT and visualized by enhanced chemiluminescence (ECL). β-tubulin was used as loading control. (**b**) The area density was evaluated by densitometry scanning and analysis with ImageJ program, the data represent the mean ± SD of three experiments performed in duplicate (significance, * *p* < 0.001 versus control sample).

**Figure 6 ijms-20-04375-f006:**
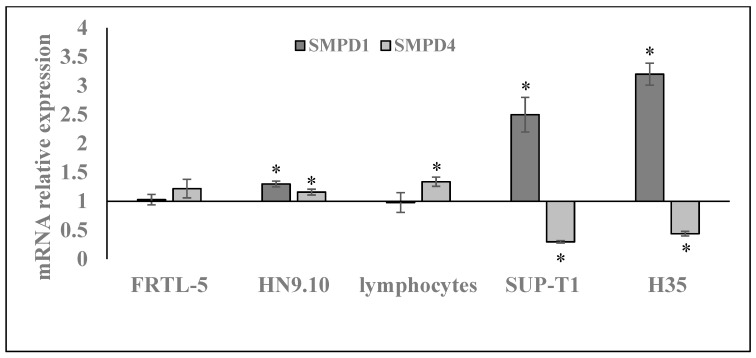
Effect of gentamicin on *SMPD1, SMPD4* gene expression in thyrocytes (FRTL-5 cells), embryonic hippocampal cells (HN9.10), human lymphocytes and non-Hodgkin’s T cell human lymphoblastic lymphoma cells (SUP-T1), hepatoma cells (H35), and human gastric cancer cells (NCI-N87). *GAPDH* expression was used as a housekeeping gene. RT-PCR analysis was performed in control and experimental cells collected after 24 h after 1.5mM GM treatment. Data are expressed as the mean ± SD of three independent experiments performed in three PCR replicates (significance, * *p* < 0.001 versus control sample).

**Figure 7 ijms-20-04375-f007:**
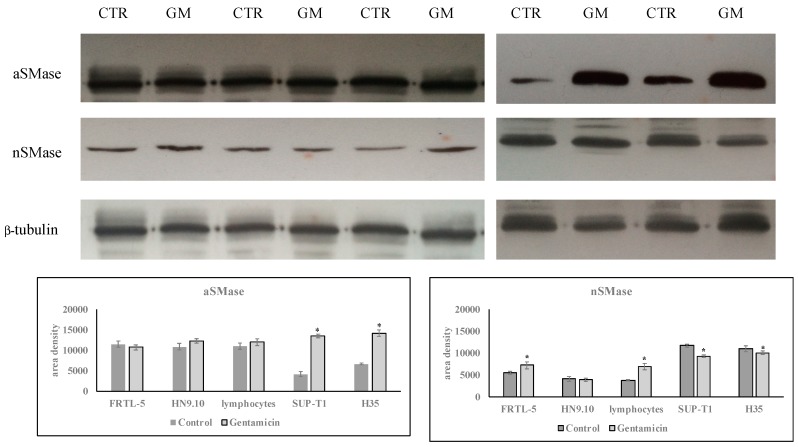
Effect of gentamicin on aSMase and nSMase protein expression. Experiments were performed in control and experimental FRTL-5, HN9.10, lymphocytes, SUP-T1, and H35 cells, collected 24 h after 1.5 mM GM treatment. (**a**) Immunoblots of proteins were probed with anti-aSMase and anti-nSMase and visualized by enhanced chemiluminescence (ECL). β-tubulin was used as loading control. (**b**) The area intensity was evaluated by densitometry scanning and analysis with ImageJ program, the data represent the mean ± SD of three experiments performed in duplicate (significance, * *p* < 0.001 versus control sample).

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
