# Peer review of "Gentamicin Targets Acid Sphingomyelinase in Cancer: The Case of the Human Gastric Cancer NCI-N87 Cells"

_ijms, 2019, doi:10.3390/ijms20184375_

Round 1

Reviewer 1 Report

In the initial review process, referees felt that this study is incomplete as they found some inadequacies in the content of the manuscript. I found that revised manuscript was improved as they added some new experimental data and some descriptions in the text were clarified. However, authors failed to provide sufficient revision regarding key issues raised by reviewers. For example, this manuscript focused on gastric cancer, however, it still used only one gastric cancer cell line. Additionally, both referees felt that it is unclear whether observed cell growth inhibition was owing to increased acid sphingomyelinase (aSMase) expression. Referee 2 suggested that they test effect of aSMase knockdown by siRNA transfection. However, authors failed to do it. In conclusion, I would like to encourage authors to perform more works to increase quality and impact of this study.

Specific comments

Authors added new mRNA data in Figure 6 but protein data should also be presented.

Author Response

In the initial review process, referees felt that this study is incomplete as they found some inadequacies in the content of the manuscript. I found that revised manuscript was improved as they added some new experimental data and some descriptions in the text were clarified. However, authors failed to provide sufficient revision regarding key issues raised by reviewers. For example, this manuscript focused on gastric cancer, however, it still used only one gastric cancer cell line.

You are right but in the first revision other tumor lines were required and we included the Lymphoblastic lymphoma cells (line SUP-T1) and the Hepatoma cells (line H35) just to demonstrate the effect of gentamicin in very distant cell lines and we reviewed all the paper based on the results obtained. We currently have no other gastric cancer cell lines in culture; so ,based on your request, we changed the title, the results and discussion

Additionally, both referees felt that it is unclear whether observed cell growth inhibition was owing to increased acid sphingomyelinase (aSMase) expression. Referee 2 suggested that they test effect of aSMase knockdown by siRNA transfection. However, authors failed to do it.

You are right but as I wrote in my previous letter that aSMase knockdownis specific of a cellular model that reproduces Niemann-Pich disease in vitro. The lack of the enzyme causes an accumulation of sphingomyelin that induces cellular degeneration and death with a great experimental variability. This is why in the last few years, we, but also other researchers, have preferred to work in vivo. (see our recent publication in IJMS, Conte C et al., 2019). Therefore the experiment is expensive and does not give easily repeatability of the data. Furthermore the accumulation of sphingomyelin creates damage and it is not easy to establish the effect of a drug in these cells. It has been reported in the discussion.

In conclusion, I would like to encourage authors to perform more works to increase quality and impact of this study.

Thank you very much for your advice, we performed the experiment suggested in specific comments

Specific comments

Authors added new mRNA data in Figure 6 but protein data should also be presented.

It has been made (see Fig.7, results and discussion)

Reviewer 2 Report

The manuscript has been improved, while there are still concerns. 

Cell viability can also be determined by live cell percentile. In Fig. 1a, 24 hours of 1.75 and 2mM  GM treament didn't have any effect on cell proliferation, while lower doses (0.75 - 1.5 mM) reduced cell number. Is it supposed to be dose-dependent? How do authors explain it? How many times was this experiment conducted? Fig. 1c shows the apoptosis analyzed by FACS. Where is the cell cycle analysis, as it was described in the methods?

Author Response

The manuscript has been improved, while there are still concerns. 

Cell viability can also be determined by live cell percentile. In Fig. 1a, 24 hours of 1.75 and 2mM  GM treament didn't have any effect on cell proliferation, while lower doses (0.75 - 1.5 mM) reduced cell number. Is it supposed to be dose-dependent? How do authors explain it? How many times was this experiment conducted? Fig. 1c shows the apoptosis analyzed by FACS. Where is the cell cycle analysis, as it was described in the methods? 

Thank you for your observation. The response of cells to drugs is strange, often low and high doses have similar effects, completely different from medium doses and therefore often there is no straight dose response. Often high doses behave like the initial doses or once the response with low doses is obtained, this does not change with higher concentrations (Int J Mol Sci. 2015 Jan 22;16(2):2307-19. doi: 10.3390/ijms16022307; Int J Mol Sci. 2019 Jul 25;20(15). pii: E3634. doi: 10.3390/ijms20153634; Eur J Cancer. 2010 Jun;46(9):1735-43. doi: 10.1016/j.ejca.2010.03.041. Epub 2010). It has been included in the results.As reported in the figure legend, the data represent the mean ± S.D. of 3 independent experiments performed in duplicate.

Data of cell cycle have been included in the results

Round 2

Reviewer 1 Report

No further comments from my side.

This manuscript is a resubmission of an earlier submission. The following is a list of the peer review reports and author responses from that submission.

Round 1

Reviewer 1 Report

This manuscript described that gentamicin (GM) can inhibit growth of a gastric cancer cell line, NCI-N87. This associates with altered transcriptional regulation of some genes potentially responsible for cell growth inhibition. This reviewer felt that the study is preliminary and incomplete. Additionally, the study lacks novelty as effect of GM on SMase metabolism in cancer and non-cancer cells has already been reported.

Specific comments

1)      Title: Authors should not generalize phenomena observed in this study to gastric cancer cells as they used only one cell line.

2)      Introduction: It is unclear why authors focus on gastric cancer in this study.

3)      Authors failed to show normal (non-cancer) cell data. This should be compared to that of cancer cells.

4)      Data shown in Figure 2 are not convincing. They should be quantitatively evaluated.

5)      Figure 4: It seems that GM treatment increases acid SMase but decreases neutral SMase expression in NCI-N87 cells. However, it is unclear whether this altered SMase regulation correlates with observed cell growth inhibition.

6)      Did the authors test another aminoglycoside antibiotics? Is this characteristic to GM?

Reviewer 2 Report

In this manuscript, the authors studied the effect of gentamicin (GM) on human gastric cancer line. They found that gentamicin treatment suppressed cell proliferation, increased acid sphingomyelinase (aSMase), and therefore demonstrated that GM acts via aSMase.

However, several issues need to addressed or fixed.

1. The results are not strong enough to conclude that GM delays cell proliferation and aggressiveness by involving specifically aSMase. Thousands of genes or proteins could be changed by drug treatment. To elucidate whether the drug functions via a specific gene/protein, more direct evidence should be presented. For instance, in this case, overexpression of aSMase should be performed to test whether it has similar effect on cell proliferation as GM did. Or knock-down or knock-out of aSMase should be carried out to determine whether it could recover the effect of GM on cell proliferation.

2. In Figure 1a, 24 hours of 0.75-1.5mM GM treatment reduced cell number, while 1.75 and 2mM didn’t. Figure 1b showed GM decreased cell viability by MTT assay dose-dependently from 0.75 to 2mM. It is not consistent.

3. In Figure 2, quantification of Ki-67 index is required.

4. It’d be better to analyze the change of cell cycle by flow cytometry.

5. PTEN function can be impacted through multiple regulations, such as its posttranslational modifications, subcellular localization, and binding proteins. To test whether PTEN pathway is involved or not, AKT/p-AKT or S6-p-S6 can be determined.

And it should be “gentamicin” in the title, but not “gentamicinin”.